

# Bacterial and fungal communities respond differently to varying tillage depth in agricultural soils

Craig Anderson[1], Mike Beare[1], Hannah L. Buckley[2] and Gavin Lear[3]

[1] Plant and Food Research, Lincoln, New Zealand
[2] School of Science, Auckland University of Technology, Auckland, New Zealand
[3] School of Biological Sciences, University of Auckland, Auckland, New Zealand

## ABSTRACT

In arable cropping systems, reduced or conservation tillage practices are linked with improved soil quality, C retention and higher microbial biomass, but most long-term studies rarely focus on depths greater than 15 cm nor allow comparison of microbial community responses to agricultural practices. We investigated microbial community structure in a long-term field trial (12-years, Lincoln, New Zealand) established in a silt-loam soil over four depth ranges down to 30 cm. Our objectives were to investigate the degree of homogenisation of soil biological and chemical properties with depth, and to determine the main drivers of microbial community response to tillage. We hypothesised that soil microbiological responses would depend on tillage depth, observed by a homogenisation of microbial community composition within the tilled zone. Tillage treatments were mouldboard plough and disc harrow, impacting soil to ~20 and ~10 cm depth, respectively. These treatments were compared to a no-tillage treatment and two control treatments, both permanent pasture and permanent fallow. Bacterial and fungal communities collected from the site were not impacted by the spatial location of sampling across the study area but were affected by physicochemical changes associated with tillage induced soil homogenisation and plant presence. Tillage treatment effects on both species richness and composition were more evident for bacterial communities than fungal communities, and were greater at depths <15 cm. Homogenisation of soil and changing land management appears to redistribute both microbiota and nutrients deeper in the soil profile while consequences for soil biogeochemical functioning remain poorly understood.

Corresponding author
Craig Anderson,
Craig.Anderson@plantandfood.co.nz

# INTRODUCTION

Tillage alters soil porosity, distributes carbon and nitrogen throughout the soil profile, impacts microbial respiration and potentially leads to carbon loss (*Singh et al., 2010*). More stable aggregate structure in the upper surfaces of non-tilled soils is proposed to improve soil porosity and moderate evaporation, improving soil water conservation (*Busari et al., 2015*). While increasing the abundance of water storage pores (*Pagliai, Vignozzi & Pellegrini, 2004*), the lower aeration of non-tilled soils may simultaneously decrease oxygen

availability, lowering aerobic turnover in the soil and decreasing gaseous losses (*Skiba, Dijk & Ball, 2002*). Consequently, the use of no-till soil management has been promoted to land managers seeking to reduce soil carbon losses and curb greenhouse gas emissions (*Conant et al., 2007*).

The impacts of soil management on microbial diversity and functioning are still under investigation. Crop residues and root exudates are the main sources of soil C (*Gougoulias, Clark & Shaw, 2014*) with tillage distributing these C-sources deeper into the soil and altering soil structure. Tillage, therefore affects microbial access to fresh C at depth, releases previously inaccessible C and changes soil water and gas distribution thus affecting microbial metabolic rates. By contrast, no-till management restricts microbial access to fresh C (by leaving residues at the surface and in the vicinity of roots) and minimises soil disturbance, therefore impacting how soil-C will be processed. Since soil microorganisms have primary control over C flows within the soil and between the soil and atmosphere (*Balaine et al., 2016*), alterations of soil C distributions by tilling are likely to impact both microbial community composition and functioning.

The impacts of no-till management on soil C stocks are variable compared with conventional tilled systems (*Helgason et al., 2014*) where C stocks may be far higher (e.g., 1.7 times greater; *Wakindiki & Njeru, 2017*) due to surface derived plant C being incorporated into the soil. With this in mind, it is conceivable that a moderate degree of tillage or inversion tillage may aid restoration of soil C stocks at deeper levels in the soil profile. However, to confirm the restoration of soil C would first require confirmation that appropriate levels of C exist, appropriate microbial communities are present that can decompose the residues at depth and that other limiting nutrients are made available. Until recently, few studies have examined the impact of different tillage practices on soil microbial community structure, specifically at depths >15 cm (*Ceja-Navarro et al., 2010*; *Navarro-Noya et al., 2013*; *Van Groenigen et al., 2010*). Since both bacterial and fungal communities play major roles in soil organic matter cycling, we examined their composition within a long-term (12-year) trial to evaluate the effects of tillage down to a depth of 30 cm. For both communities, we hypothesised that there would be weaker depth-related gradients in community composition in tilled soil, since tillage should homogenise the soil and overshadow any depth-dependant effects. To further explore the role of tillage on depth-related gradients in soil microbial community composition, we chose to compare communities in untilled soil to communities in soil tilled to depths of either 10 or 20 cm. We also expected fungal communities to be more prone to disturbance from ploughing because of their extensive hyphal networks (*Wardle, 1995*). Therefore, our objectives were to investigate the degree of homogenisation of soil biological and chemical properties to depths of 30 cm, and to identify the main drivers of microbial community responses to tillage intensity.

## METHODS

### Experimental site and field trial description

Replicated soil samples were taken pre-harvest (09/03/2012) from 15 plots at a long-term tillage trial run by Plant & Food Research, near Lincoln, in the South Island of New Zealand

(43°40′S latitude, 172°28′E longitude; mean annual air temperature 11.4 °C, mean annual rainfall 867 mm). The soil underlying this site is a Wakanui silt loam, classified as Udic Dystocrypt according to USDA taxonomy (*Natural Resources Conservation Service, 1999*). Before trial establishment, the site was sheep-grazed, irrigated permanent pasture that had not been cultivated for at least 14 years. Three tillage methods applied in Spring and Autumn seasons were evaluated, these being; No-tillage (Nn): no cultivation, seeds direct drilled; Minimum tillage (Mm): the top 10 cm cultivated using a spring tined implement, followed by secondary cultivation (harrowing and rolling twice); Intensive tillage (Ii): cultivation to ∼20 cm using a mouldboard plough, followed by secondary cultivation (one pass with a spring tined implement followed by harrowing and rolling twice). All tillage operations were carried out using standard commercial equipment. Spring-sown main crops rotation included barley, wheat, and peas. They were followed by winter-grazed (sheep) cover crops (oats or forage brassicas). All crops were sown using a Great Plains direct drill. Fertiliser (N and P) were applied to the spring crops to ensure these nutrients were not limiting. Plots representing the original ryegrass-clover pasture were maintained within the trial as a control. To balance the trial design, these plots were split into subplots; permanent pasture (Pp), and permanent fallow (Pf). The Pp sub-plots were grazed with sheep (typically 10 times per year; 20 sheep per plot). The main fertiliser applied to the Pp plot was superphosphate. The Pf subplots received no fertiliser and had no animal or vehicle trafficking throughout the trial. Herbicide (Glyphosate) was used to maintain the Pf subplots plant free. Management (irrigation, fertiliser, grazing regime) of the Pp plots remained the same as before the trial. All treatments (i.e., Arable crops, Pp and Pf) were irrigated in summer to ensure that water was not limiting to pasture or crop growth. Treatment plots were replicated three times in an incomplete Latin square (i.e., five treatments × three replicate plots = 15 plots; see Fig. 1). The size of individual plots was 28 m × 9 m. Further trial details can be obtained from *Fraser et al. (2013)*. The long term field trial was operated by Plant and Food Research. No additional permits were required for sample collection.

Two types of soil samples were taken from each plot: (1) six surface 25 mm diameter core samples (0–7.5 cm) where each sample was analysed separately to confirm the impact of spatial heterogeneity on sample data and (2) eight deeper 5 cm diameter core samples separated into four depth ranges (0–7.5 cm, 7.5–15 cm, 15–25 cm and 25–30 cm), which were later composited by depth (Fig. 1). Soil used for chemical and physical analysis was stored at 4 °C prior to use and 2 g aliquots of each sample frozen in Eppendorf tubes for DNA extraction. Soil subsamples were taken from each depth to measure: (1) water content, (2) pH, (3) bulk density/mean weight diameter (MWD), (4) exchangeable acidity, (5) exchangeable aluminium, (6) concentrations of C and N, and (7) microbial biomass C and N.

## Soil chemical analysis

Gravimetric soil moisture content was determined by the mass difference before and after drying at 105 °C for 16 h. The pH of each sample was determined using a glass electrode at 1:2 field moist sample to water ratio (*Hendershot, Lalande & Duquette, 2008*). Bulk density
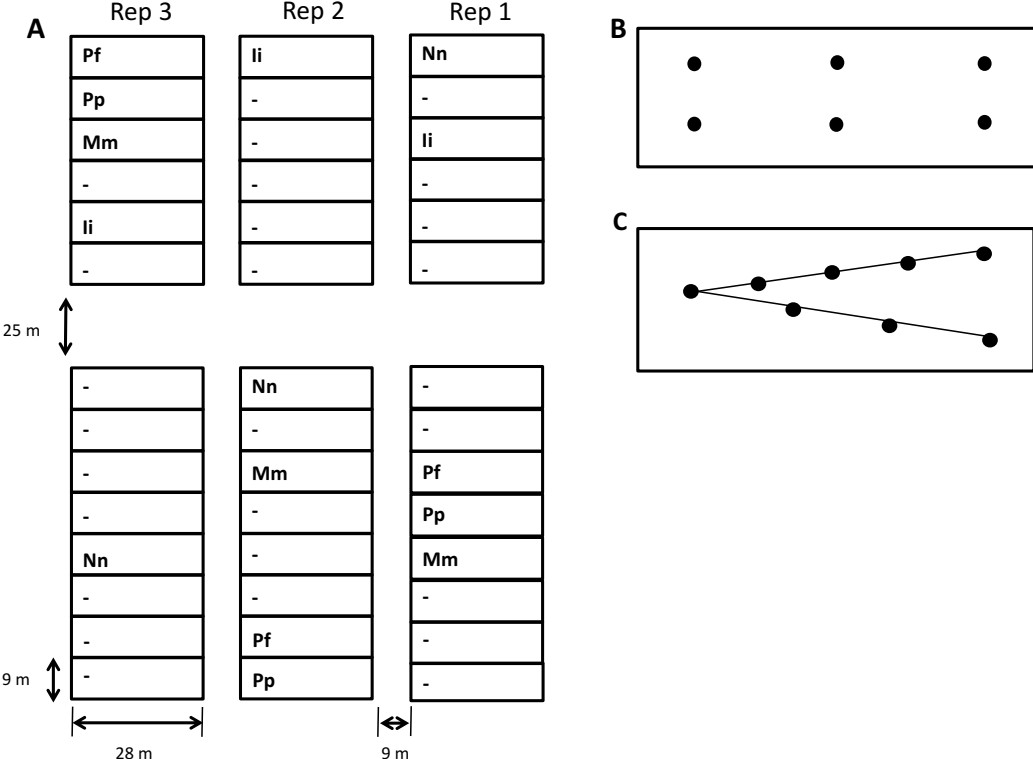

**Figure 1** **Map of the field plot trial (located: Lat. 43°40′03″ S, Long. 172°28′05″ E) and soil sampling strategy.** (A) Map of study site. Tillage treatments are (Pp) permanent pasture, (Pf) permanent fallow, (Ii) intensive tillage to 20 cm, (Mm) minimum tillage to 10 cm and (Nn) no-till. Plots labelled (-) represent a variety of treatments not investigated in the present study. All 15 plots (5 treatments × 3 replicates) were sampled twice on the same day. (B) During the first sampling event six samples were collected from each plot from a depth of 0–7.5 cm to provide a total of 90 samples. (C) During the second sampling event, eight sample cores were collected from each plot and cores separated into depths of 0–7.5 cm, 7.5–15 cm, 150–25 cm and 25–30 cm, before the soil from each depth was composited, providing an additional 60 samples for analysis.

(<4 mm) was calculated from the weight of field-moist soil of known volume, corrected for its stone and moisture contents. Aggregate stability or mean weight diameter (MWD) was determined by first separating 2–4 mm aggregates from whole soil by sieving, and then air-drying them at 25 °C before aggregate stability determination using a wet-sieving method (*Kemper & Rosenau, 1986*). The air-dried 2–4 mm aggregates (50 g) were sieved underwater for 20 min on a nest of sieves (2.0, 1.0 and 0.5 mm diameter). The soil remaining on each sieve was weighed after oven drying at 105 °C. The aggregate stability was expressed as a mean weight diameter (MWD):

$$MWD = \sum_{i=1}^{n} x_i w_i$$

where $x_i$ is the mean diameter of adjacent sieves and $w_i$ is the proportion of the total sample retained on a sieve.

Exchangeable acidity (Exch. Acid.) and aluminium (Exch. Al.) was determined by extraction using 1 $M$ KCl. The amount of $H^+$ and $Al^{3+}$ in the extracts was determined by titration as described by *Sims (1996)*. Total carbon (C) and nitrogen (N) contents were determined by the Dumas dry combustion method at 950 °C using a Truspec C/N analyzer (LECO, St. Joseph, MI, USA).

Microbial biomass C (MBC) and N (MBN) were determined by chloroform fumigation-extraction as described by *Sparling & West (1988)*. Pre- and post-fumigation extracts were analysed for organic C by combustion catalytic oxidation using a TOC-V$_{CSH}$ analyzer (Shimadzu Corporation, Kyoto, Japan) and for organic N by the persulfate oxidation method described by *Cabrera & Beare (1993)*. Physicochemical data collected from the site are provided in Supplemental Information 1.

## Production and manipulation of ARISA data from extracted DNA
### Automated Ribosomal Intergenic Spacer Analysis (ARISA)

For each sample (160 in total), DNA was extracted from 0.25 g freeze-dried soil using Powersoil®-htp 96 well DNA isolation kits (MoBio Laboratories Inc., Carlsbad, CA, USA) following the manufacturer's instructions. Automated Ribosomal Integenic Spacer Analysis (ARISA) was then used to evaluate the composition of bacterial and fungal communities in each sample according the method of *Lear et al. (2008)*. This PCR-based method characterises the structure of the microbial community within each sample by recording the length (in base pairs, b.p.) of the intergenic spacer (ITS) regions of the constituent microbes, i.e., between the bacterial 16S rRNA and 23S rRNA genes or the fungal 18S rRNA and large ribosomal subunit genes.

PCR amplification of bacterial ITS regions was undertaken on the extracted DNA using Promega GoTaq® Green DNA polymerase master mix (Invitro Technologies Ltd., Auckland, New Zealand) and the primers SDBact (5′ -TGC GGC TGG ATC CCC TCC TT-3′ ) and LDBact (5′ -CCG GGT TTC CCC ATT CGG) (*Ranjard et al., 2001*), with the following amplification conditions: (i) 95 °C for 5 min; (ii) 30 cycles of 95 °C for 30 s, 61.5 °C for 30 s, 72 °C for 90 s and then (iii) 72 °C for 10 min. The primer SDBact was labelled at the 5′ -end with HEX (6-carboxyhexafluorescein) fluorochrome (Invitrogen Molecular Probes, New Zealand) to enable analysis by ARISA.

For the fungi, the PCR primers used were FunNS1 (5′- GAT TGA ATG GCT TAG TGA GG −3′) (*Martin & Rygiewicz, 2005*) and 3126T (5′ - ATA TGC TTA AGT TCA GCG GGT −3′) (*Ranjard et al., 2001*). PCR amplification used the Phusion® polymerase (NEB, Ipswich, MA, USA) according to the manufacturer's instructions, with the following amplification conditions: (i) 98 °C for 2 min; (ii) 35 cycles of 98 °C for 10 s, 55 °C for 30 s, 72 °C for 45 s and then (iii) 72 °C for 20 min. The primer FunNS1 was labelled at the 5′-end with FAM (6-carboxyfluorescein) fluorochrome (IDT, Asia Pacific, Singapore).

Products were each purified (Zymo DNA clean and Concentrator kit; Ngaio Diagnostics Ltd., Nelson, New Zealand) and DNA concentration (ng $\mu l^{-1}$) individually determined using a Nanodrop ND-100 spectrophotometer (NanoDrop Technologies, Rockland, DE, USA). Appropriate volumes of cleaned PCR product (diluted with ultrapure $H_2O$ if necessary) providing a final DNA mass of 5 to 10 ng were then transferred to a 96-well
sequencing plate and dried in a speedvac for 2 h at 60 °C. The dry sample was resuspended in 15 μl Hi-Di deionised formamide and Genescan LIZ-1200 internal size standard (ABI Ltd.). The sample was heated (5 min, 95 °C) and analysis was carried out on an ABI 3130XL genetic analyser with POP7 chemistry and a 36 cm array (ABI Ltd.).

## Quantitative analysis of ARISA data

GENEMAPPER software (v. 3.7; ABI Ltd) was used to assign a fragment length (in nucleotide base pairs) to ARISA peaks, via comparison with the standard ladder (LIZ1200; ABI Ltd.). To include the maximum number of peaks whilst excluding background fluorescence, only peaks with a fluorescence value of 50 U or greater were analysed. As the 16S-23S region is thought to range between ∼140 and 1530 bp (*Fisher & Triplett, 1999*), fragments <150 bp were excluded from analysis. No samples contained fragments >1,000 bp. The same size (bp) parameters were used for the fungi as these samples also did not contain any fragments >1,000 bp. The total area under the curve was normalised (to 100) to remove differences in profiles caused by different initial DNA template quantities, and peak size was rounded to the nearest whole number. Each ARISA sample therefore consisted of 851 operational taxonomic groupings of bacteria or fungi, which represent the length of the intergenic spacer region of constituent microbes (in bp), thereby providing an informative profile of the bacterial and fungal community composition within each sample. OTU tables are available provided in Supplemental Information 2–3.

To visualise multivariate patterns in the soil microbial community structure among samples, nonmetric multidimensional scaling (nMDS) was done using the Bray Curtis measure. Rather than using multivariate analysis of variance MANOVA to test the data, which assumes normal distributions, and implicitly Euclidean distances, we chose to use permutational MANOVA (or PERMANOVA; *Anderson, 2001*) with the data assigned to the factors Treatment (Pf, Pf, Ii, Nn and Mm) and Depth (0.7.5, 7.5–15, 15–25 and 35–30 cm). MVDISP was used to compare the extent of multivariate data dispersion across these groups. These multivariate analyses were performed using the PRIMER v.6 computer program (*Clarke & Gorley, 2006*) with the additional add-on package PERMANOVA + (*Anderson, Gorley & Clarke, 2008*).

We used the aov function in R version 2.14 (*R Core Team, 2012*) to perform analyses of variance on soil chemical data using a two-way layout (treatment; depth), with interaction terms. Canonical redundancy analysis (RDA) and was used to summarise variation in the bacterial and fungal community data that could be explained by our set of explanatory variables (e.g., pH, soil water content). Variance partitioning was then performed using the function varpart.MEM in R, following *Borcard, Gillert & Legendre (2011)* to describe and partition variation in community composition between two sets of explanatory variables: soil chemical properties and geographic location.

## RESULTS AND DISCUSSION

Analysis of the surface soil samples (0–7.5 cm) showed significant variation in microbial community composition among treatments (Fig. 2, PERMANOVA all $P < 0.001$). Bacterial and fungal composition from the five treatments differed significantly irrespective of

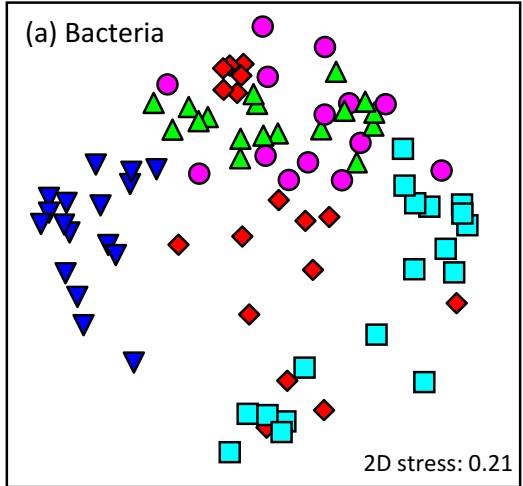
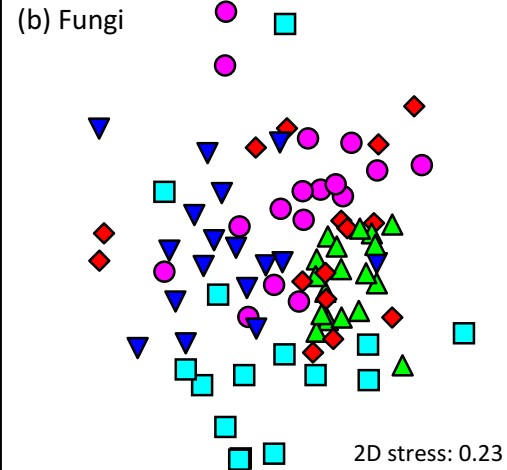

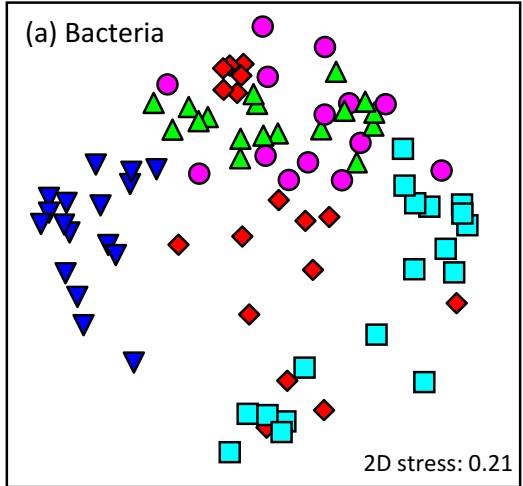 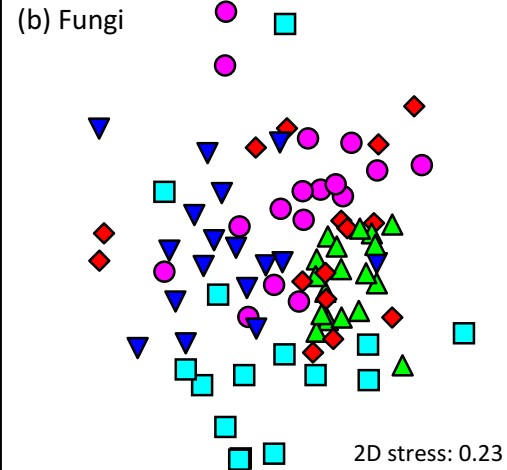

▼ Permanent pasture, no-till
□ Permanent fallow, no-till
◆ Cropping, intensive till
▲ Cropping, moderate till
● Cropping, no-till

**Figure 2** **Non-metric multi-dimensional scaling plots of (A) bacteria; and, (B) fungi grouped according to treatments.** Impact of tillage treatment on soil microbial community composition. Plots are non-metric multi-dimensional scaling plots of (A) bacterial; and, (B) fungal community data grouped according to treatments: (triangle-down) Permanent pasture, (square) Permanent fallow, (diamond) Intensive tillage, (triangle-up) Moderate tillage, (circle) No-till. The scaling is based on a Bray-Curtis similarity matrix of ARISA profiles. All data are from soil samples of 0–7.5 cm depth. 2D stress values are 0.21 and 0.23 for bacterial and fungal data, respectively. PERMANOVA revealed significant treatment effects for both bacterial and fungal communities ($p < 0.001$).

whether the data remained untransformed or was $\log(X + 1)$ transformed to remove computational bias derived from dominant OTUs (operational taxonomic units—broadly representing 'unknown' phyla). These results confirm soil management practices impact the composition of both bacterial and fungal communities, supporting the findings of other recent studies (*Busari et al., 2015*; *Ceja-Navarro et al., 2010*; *Mathew et al., 2012*).

The treatment differences among the bacterial community data were more pronounced than those of the fungal data, as the former formed more distinct clusters on an nMDS plot (Fig. 2). The removal of the Pp and Pf data from the analysis further improved the separation of the cropped tillage treatments and reduced the 2D-Stress goodness of fit statistic to 0.16 and 0.15 for bacteria and fungi respectively, improving the certainty of the visual nMDS solution (*Cox & Cox, 1992*). All pairwise PERMANOVA comparisons among treatments were significant for bacteria, but only three (Pp-Ii, Pf-Nn, and Mm-Pf) were significant for fungi (i.e., PERMANOVA *P* all <0.001) suggesting that bacterial communities in the surface soil (0–7.5 cm) were more sensitive to tillage treatment than fungal communities. This tillage treatment effect may be because bacteria tend to dominate in soils that are

Table 1  **Mean values (±S.E.) for soil chemistry variables and significance from two-way ANOVA ($N$ = 45 for all comparisons; soil chemistry data were not generated for 25–30 cm) samples.** Different letters (a, b, c, d) indicate significantly different treatment effects using Tukey's Honestly significant difference multiple comparison tests. Treatments are as follows: Pp, Permanent pasture; Pf, Permanent fallow; Ii, Intensive tillage; Mm, Moderate tillage; Nn, No-till. Depths are as follows: t, top (0–7.5 cm), m, middle (7.5–15 cm) and b, bottom (15–25 cm).

| Variable | Unit | Mean | ±S.E. | Treatment $P$ | Depth $P$ | Interaction $P$ | Treatment rank[a] | Depth rank[a] |
|---|---|---|---|---|---|---|---|---|
| Total C | g kg$^{-1}$ | 21.9 | ±0.09 | <0.001 | <0.001 | <0.001 | $^a$Pp>$^b$Mm>$^b$Nn>$^b$Ii>$^c$Pf | $^a$t>$^b$m>$^c$b |
| Total N | g kg$^{-1}$ | 1.90 | ±0.01 | <0.001 | <0.001 | <0.001 | $^a$Pp>$^b$Mm>$^b$Nn>$^b$Ii>$^c$Pf | $^a$t>$^b$m>$^c$b |
| MWD | mm | 1.30 | ±0.09 | <0.001 | <0.001 | 0.015 | $^a$Pp>$^b$Nn>$^b$Mm>$^c$Ii>$^d$Pf | $^a$t>$^b$m>$^c$b |
| MBC | μg g$^{-1}$ | 386.91 | ±32.9 | <0.001 | <0.001 | <0.001 | $^a$Pp>$^b$Mm>$^b$Ii>$^b$Nn>$^c$Pf | $^a$t>$^b$m>$^c$b |
| MBN | μg g$^{-1}$ | 59.17 | ±4.70 | <0.001 | <0.001 | <0.001 | $^a$Pp>$^b$Mm>$^b$Ii>$^b$Nn>$^c$Pf | $^a$t>$^b$m>$^c$b |
| pH | | 5.43 | ±0.06 | <0.001 | <0.001 | 0.003 | $^a$Pp>$^b$Mm>$^b$Nn>$^b$Ii>$^c$Pf | $^a$m>$^a$t>$^b$b |
| Moisture | % | 21.03 | ±0.44 | <0.001 | <0.001 | <0.001 | $^a$Nn>$^a$Ii>$^a$Mm>$^a$Pf>$^b$Pp | $^a$t>$^b$m>$^c$b |
| Exch. acid | cmol$_c$kg$^{-1}$ | 0.53 | ±0.05 | <0.001 | 0.822 | 0.217 | $^a$Pf>$^{bc}$Ii>$^{bc}$Nn>$^{bc}$Mm>$^c$Pp | $^a$b>$^a$m>$^a$t |
| Exch. al | cmol$_c$kg$^{-1}$ | 0.34 | ±0.04 | <0.001 | 0.267 | 0.178 | $^a$Pf>$^{bc}$Ii>$^{bc}$Nn>$^{bc}$Mm>$^c$Pp | $^a$b>$^a$m>$^a$t |

**Notes.**
[a]Means for each level of treatment and depth were ranked from highest to lowest.

intensively managed, where they drive decomposition and nutrient cycles (*Garcia-Orenes et al., 2013*). The greatest pairwise Bray–Curtis distances between data representing any two treatments for both taxa were between the various no-till treatments, i.e., between Pf and Pp for bacteria and between Pf and Nn for fungi. Since greater average Bray–Curtis distances among data indicate greater differences in overall community composition, these findings confirm that tillage as a disturbance drives microbial community composition to a lesser degree than other management effects, such as the presence of permanent pasture, grazing or vegetation removal. We suggest tillage treatment differences have less effect because for both bacterial and fungal communities, average community similarity (i.e., average Bray-Curtis distances were least) comparing the treatments Mm and Nn. For samples taken across depths, two-way ANOVA showed that all soil chemical properties, except concentrations of exchangeable aluminium and acidity, differed significantly by treatment depth (Table 1 and Fig. S1). With the exception of soil water content, the greatest difference among treatments was again between the non-till control treatments Pp and Pf, and the biggest difference among depths was between 0–7.5 cm and 15–25 cm, noting that chemical data was never obtained from the deepest (25–30 cm) samples. Previous research at this field site has indicated that crop productivity is not influenced by tillage, neither is nutrient input (D Curtin, pers. comm., 2017). However, tillage introduced a degree of homogenisation that was reflected in the depth profiles of soil chemical properties and nutrient distribution. Soil chemical attributes varied little with depth under intensive tillage, which is of relevance since variation in nutrient inputs and soil depth are suggested to be important drivers of microbial community change (*Jangid et al., 2008*; *Jeffery et al., 2007*).

NMDS and PERMANOVA showed that bacterial community composition, like soil chemical properties, varied predictably with depth and tillage treatment. Treatment effects were greatest among the shallowest soil samples (≤15 cm) compared with deeper soil, with these data points being separated further apart on the nMDS plot as compared to samples collected at greater depth (Fig. 3A). Multivariate dispersion index values (i.e., mean

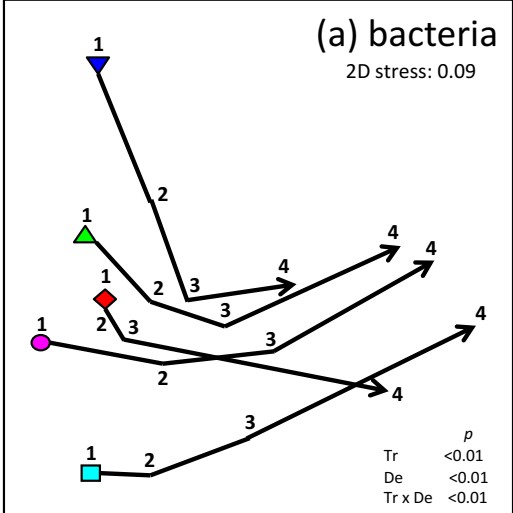
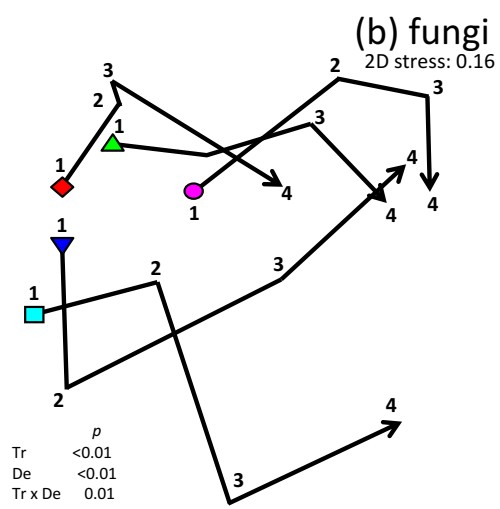

**Figure 3** **Non-metric multi-dimensional scaling plots of (A) bacteria; and, (B) fungi grouped according to treatment and sampling depth.** Impact of crop management on microbial community composition measured at different soil depths. Plots are non-metric multi-dimensional scaling plots of (A) bacterial; and, (B) fungal community data grouped according to treatments (triangle-down) Permanent pasture, (square) Permanent fallow, (diamond) Intensive tillage, (triangle-up) Moderate tillage, (circle) No-till. The scaling is based on a Bray–Curtis similarity matrix of ARISA profiles. The trajectory shows the movement of data points related to depth (1) 0–7.5 cm, (2) 7.5–15 cm, (3) 15–25 cm, (4) 25–30 cm for average data from each treatment. 2D stress values are 0.09 and 0.16 for bacterial and fungal data, respectively. The significance (PERMANOVA $p$ values) of differences related to treatment (Tr), sample depth (De) and their interaction (Tr $\times$ De) are shown on each plot.

Bray Curtis dissimilarities among samples within groups) confirmed greater variation in bacterial community composition comparing samples collected at shallower depth (MVDISP = 1.3, 1.2, 0.8 and 0.8 for samples collected from 0–7.5, 7.5–15, 15–25 and 25–30 cm, respectively, where greater values indicate greater multivariate data dispersion within the group). Overall, PERMANOVA only showed significant pairwise treatment effects to a depth of 25 cm. These results are consistent with our hypothesis that the tillage effects on microbial communities would decline or weaken with depth.

Fungal community composition changed with increasing sample depth but unlike the bacterial community data, no consistent pattern is observable aside from some separation between sample data from tillage treatments that were cropped versus non-cropped Pp and Pf treatments (Fig. 3B). MVDISP calculations showed that the multivariate dispersion of samples was lowest for those taken at 0–7.5 or 25–30 cm and therefore the fungal communities did not show the same patterns of decreasing variation among treatments with depth (MVDISP = 0.8, 1.1, 1.5 and 0.7 for samples collected from 0–7.5, 7.5–15,
15–25 and 25–30 cm, respectively). A number of reasons can be proposed to explain this finding. First, since most soil-dwelling fungi are aerobic (*Gruninger et al., 2014*) is it commonly observed that they form weak depth related gradients in composition compared to bacteria, which have a far greater diversity of metabolic traits related to respiration (*Richardson, 2000*). Additionally, being larger organisms, the biomass of single multicellular fungi is likely to be represented at multiple soil depths (*Genney, Anderson & Alexander, 2006*), thereby exhibiting weaker depth-related gradients in composition across small spatial scales. However, it remains possible that the apparent difference in bacterial and fungal community treatment responses is also impacted by the choice of DNA fragments amplified, which can influence both the number and composition of OTUs detected in a community (*Kumar et al., 2011*). To address this issue, it may be desirable in future studies to assess variation in both bacterial and fungal community composition using a range of genetic markers, analysed by either DNA fingerprinting (*Adair, Wratten & Lear, 2013*) or sequencing methods (*Hermans et al., 2017*).

Canonical redundancy analysis (RDA) was used to describe and partition variation in community composition between two sets of explanatory variables: soil chemical properties and geographic location. Soil chemical properties alone explained 25% and 22% of the variation in bacterial and fungal community composition, respectively; whereas, spatial location could explain only 2% or 3% of the variation in bacterial or fungal composition, respectively. This confirms that within this field trial microbial communities are responding more to soil chemical properties rather than to spatial location in the field or plot.

Two-way ANOVA confirmed that relative bacterial OTU richness (variety or number of OTUs) in the 0–7.5 cm depth was greater than at lower depths (Fig. 4; $P < 0.001$), but did not differ among treatments. In contrast, fungal richness did not significantly differ by depth or treatment perhaps also explaining why variance partitioning showed that soil chemical properties explained 31% of the bacterial richness but only 5% of the variation in fungal richness. Depth $\times$ treatment interactions were not significant for the OTU richness of either taxon. However, the general lack of effect of either sample depth or treatment on microbial community richness was not unexpected. DNA fingerprinting methods do not provide species level diversity estimates and are not suitable to report absolute measures of community richness (*Fierer, 2007*). Additionally, the metabolic complexity of microbial life means that high levels of diversity are commonly observed even in environments that are commonly perceived as being extreme for life, such as high temperature, highly acidic and polluted environments (*Du, Ren & Hu, 2009*; *Savage et al., 2016*), or in deep sediments for example (*Lehman et al., 2001*).

Overall, our study confirms tillage has significant impacts for both the biology and chemistry of soil. Previous studies examining soil carbon and nitrogen concentrations have suggested no significant difference, or even lower soil carbon concentrations under reduced tillage systems (*Powleson et al., 2014*). However, tillage practices have the potential to impact not only total carbon and nitrogen stocks, but their distribution in the soil. Here, as observed by others (*Du, Ren & Hu, 2009*; *Zhao et al., 2015*), we confirm concentrations of soil and microbial biomass carbon and nitrogen were reduced in the surface soil by tillage, whereas they were greater at depth, indicating the transfer of biomass to lower soil layers

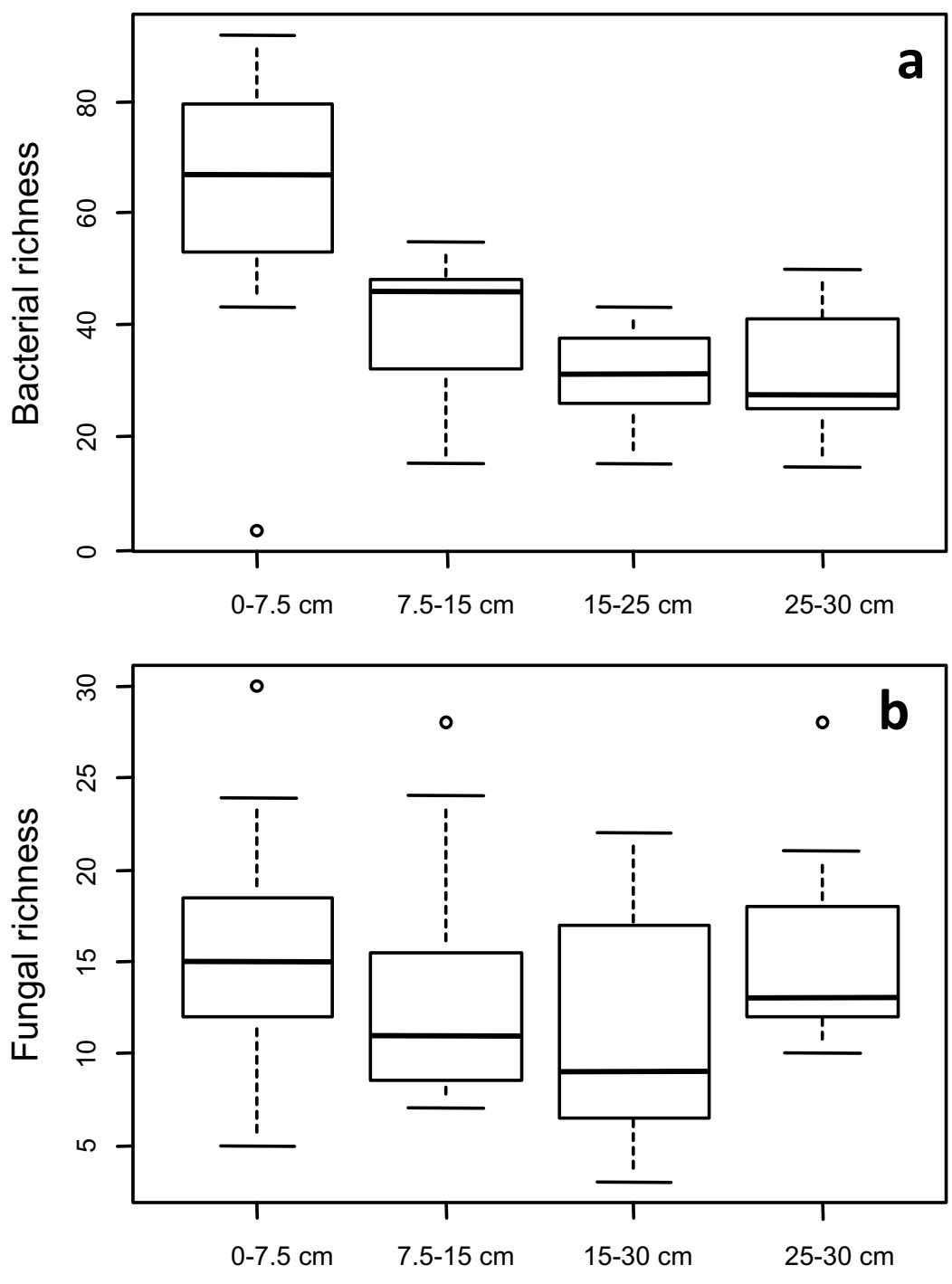

**Figure 4   Taxon richness of (A) bacteria and (B) fungi as a function of soil depth.** Median values are represented by a thick line in each box, and whiskers represent 1.5× the interquartile range.

by mechanical tillage. Although the ARISA methodology lacks in-depth precision below perhaps order level (*Gobet, Boetius & Ramette, 2014*), the method was sufficient to indicate that bacterial community composition is more responsive to tillage treatment differences than fungi. Tillage affected the composition, but not the richness, of soil microbial communities. Changes in community composition with depth appeared to be related to tillage intensity with the deeper mouldboard plough (0–20 cm) acting to homogenise soil nutrients and microbial communities throughout the soil depth affected by this disturbance. Moderate tillage with disc harrow (0–10 cm) and the no-till treatments behaved similarly to each other, exhibiting a higher degree of community variation with depth. We confirm the significant impact of tillage on soil microbial community composition, perhaps resulting from the homogenisation of local soil chemical characteristics. Soil microorganisms are known to impact agricultural production, for example by controlling nutrient availability and by mediation of plant stress tolerance (*Souza, Ambrosini & Passaglia, 2015*; *Ferrara et al., 2012*; *Zahran, 1999*). Further studies, perhaps also investigating plant biomass yield and quality, are now required to confirm the impact of tillage related changes in soil microbial community composition for plant health and production potential.

### Funding
This work was funded by the New Zealand Institute for Plant and Food Research. The funders had no role in study design, data collection and analysis, decision to publish, or preparation of the manuscript.

### Grant Disclosures
The following grant information was disclosed by the authors:
New Zealand Institute for Plant and Food Research.

### Competing Interests
Hannah L. Buckley is an Academic Editor for PeerJ. Craig Anderson and Mike Beare are employees of Plant and Food Research, Lincoln, New Zealand.

### Author Contributions
- Craig Anderson and Gavin Lear conceived and designed the experiments, performed the experiments, analyzed the data, contributed reagents/materials/analysis tools, wrote the paper, prepared figures and/or tables, reviewed drafts of the paper.
- Mike Beare conceived and designed the experiments, performed the experiments, contributed reagents/materials/analysis tools, wrote the paper, reviewed drafts of the paper.
- Hannah L. Buckley analyzed the data, wrote the paper, prepared figures and/or tables, reviewed drafts of the paper.

### Data Availability
The raw data has been provided in the Supplemental Files.

## Supplemental Information

Supplemental information for this article can be found online at http://dx.doi.org/10.7717/peerj.3930#supplemental-information.

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
