# Peer review of "Bacterial and fungal communities respond differently to varying tillage depth in agricultural soils"

_PeerJ, doi:10.7717/peerj.3930_

## Round 0.1 · original submission · Major Revisions

· Academic Editor

Major Revisions

Two reviewers have evaluated the manuscript and both recommended a better statistical analysis and a more focussed discussion.

Reviewer 1 ·

Basic reporting

Most of the for the Basic reporting season is acceptable. However, below are some comments about the English language.
Line 50 change Tillage therefore for Tillage, therefore,
Experimental design
Line 62 This however … The use of "this" brings an unclear antecedent. I suggest that the authors rewrite the sentence making a better connection with the previous statement.
Line 83 change Prior to for Before
225 change This for This tillage treatment effect
Line 229 and line 231 the use of this brings an unclear antecedent.
Line 304 change eachother for each other

Experimental design

Some parts of the topic experimental design need to be better explained. Please see the questions and comments below.

The authors proposed the nMDS analysis on their material and methods (see line 194). However, during their results and discussion, they used the term MDS, which is a different kind of multivariate analysis. Thus, I think there is a mistake on the use of MDS term and the authors should do a double check on their manuscript.

For the lines 227-229 and 231 and 232, I have a doubt about which are the statistical analyses that support the both statements. My suggestion is to rewrite these sentences and citing the analysis which endorses it.

The authors used the two-way ANOVA analysis ( e.g. line 233), however, they did not describe in methodology.

I suggest that the author use the envif analysis from Vegan package in R (http://cc.oulu.fi/~jarioksa/opetus/metodi/vegantutor.pdf) to evaluate the correlation between the environmental table (table 1) and the nMDS results. This analysis may contribute to a better visualization of which are the chemical and physical attributes that better explain the nMDS results, being helpful to confirm the hypotheses proposed .

My suggestion for the lines 296-298 is to include a reference for the ARISA methodology statement.

Validity of the findings

no comment

Comments for the author

I commend the authors for the manuscript, evaluating the effects of tillage practices and depths on soil microbial community. The article is written in a good scientific English and text structure. If there is a weakness, it is in the statistical analysis and some better explanation about some topics during the text ( as I have noted above). In my opinion, after the answer of some question and the inclusion of some analysis that could enrich the discussion, the manuscript will be ready the Acceptance.

Reviewer 2 ·

Basic reporting

The basic reporting meet the criteria, except for:

-The tittles of the figures would benefit from being more descriptive. For example, Figure 2 and three have identical titles.

Experimental design

The experimental aspects appear sound.

Since different tillage techniques are used, it seems relevant to justify the reason for studying them both in relation to depth.

Validity of the findings

I have some comments regarding the validity and discussion of the findings.

The authors could discuss the potential for the fungal community results to have been influenced by methodological factors in the molecular approach. It seems the relative error is greater and thus it appears that the sensitivity for the bacterial and fungal analyses may not be comparable? Thus the conclusion of who is more or less responsive to the management, may need to be revised.

The discussion in lines 284-293 seems too general and not sufficiently insightful and focused.

The conclusion section seems to be too broad, not focused enough of the main findings and their implications.

---

## Round 0.2 · Minor Revisions

· Academic Editor

Minor Revisions

The authors have revised the manuscript accordingly. However there are still a few snags, especially in the statistical analysis and interpretation. Further clarification of the methods will be beneficial.

For example, Line 215 reads "This variation indicates that sampling was at the appropriate scale for determining among-treatment differences." Statistical significance does not mean that the sampling is adequate. There is no "cause and effect" relation, I suggest removing that sentence.

What is permutational multivariate analysis of variance? Why is it needed?

What is 2D stress value? What does it mean

L. 207 what is the purpose of RDA?

A brief explanation would make the paper more reader-friendly.
L 244 soil water content

---

## Round 0.3 · accepted · Accept

· Academic Editor

Accept

Looking forward to the publication of this paper.